# Cross-Sim-NGF: FFT-Based Global Rigid Multimodal Alignment of Image Volumes using Normalized Gradient Fields

Johan Öfverstedt[1][0000−0003−0253−9037], Joakim Lindblad[1][0000−0001−7312−8222], and Nataša Sladoje[1][0000−0002−6041−6310]

Department of Information Technology, Uppsala University, Uppsala, Sweden

**Abstract.** Multimodal image alignment involves finding spatial correspondences between volumes varying in appearance and structure. Automated alignment methods are often based on local optimization that can be highly sensitive to initialization. We propose a novel efficient algorithm for computing similarity of normalized gradient fields (NGF) in the frequency domain, which we globally optimize to achieve rigid multimodal 3D image alignment. We validate the method experimentally on a dataset comprised of 20 brain volumes acquired in four modalities (T1w, Flair, CT, [18F] FDG PET), synthetically displaced with known transformations. The proposed method exhibits excellent performance on all six possible modality combinations and outperforms the four considered reference methods by a large margin. An important advantage of the method is its speed; global rigid alignment of 3.4 Mvoxel volumes requires approximately 40 seconds of computation, and the proposed algorithm outperforms a direct algorithm for the same task by more than three orders of magnitude. Open-source code is provided.

**Keywords:** Image registration · global · exhaustive search · NGF · FFT · matching · GPU implementation.

## 1 Introduction

Multimodal image alignment (also known as registration) involves finding correspondences between images with varying degrees of difference of appearance and structure [18], often applied with the goal of combining the complementary information of each modality via image fusion. Alignment of large displacements is particularly challenging since correspondences to be inferred are far apart and presence of multiple local optima becomes increasingly problematic as the search space grows, thereby often requiring global contextual and spatial information.

A large number of methods exist for multimodal alignment [14], including local optimization methods based on mutual information (MI) [17,8] or normalized gradient fields (NGF) [13,3], and representation extraction techniques based on local self-similarities [4] or Deep Feature Learning [12,5]. Most of the (intensity-based) methods are based on some form of local optimization, which

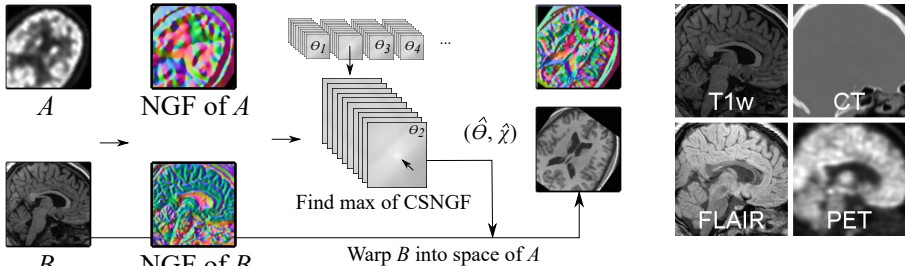

(a) Illustration of the global image volume alignment method.

(b) The modalities included in this study.

Fig. 1: Main steps of one level of the multi-level rigid alignment method (a), and examples of the modalities considered in the evaluation (b) (images from [9]). (a) Two image volumes of modalities $A$ (here [18F] FDG PET), and $B$ (T1 weighted MR), are used as input. For a set of 3D rotations $\boldsymbol{\theta}$, the similarity measure $s_{\mathrm{ANGF}}$ between the NGF of $A$ and the NGF of $B$ (rotated), (here shown as RGB images where each color channel represents one component of the 3D vector field $n_1(\cdot; A), n_2(\cdot; A), n_3(\cdot; A)$) is computed for all 3D displacements. The rigid alignment $(\hat{\boldsymbol{\theta}}, \hat{\chi})$ is found by locating the maximum $s_{\mathrm{ANGF}}$.

usually require a good initial guess to work well. However, several global alignment methods do exist, including [6,1] as well as a recently proposed method based on the cross-mutual information function (CMIF) [10].

We propose a new global alignment method based on NGF that is fast and exhibits excellent performance on a rigid multimodal 3D medical image alignment task. Our evaluation on 6 pairs of modality combinations shows that it outperforms well known methods which rely on local optimization of MI [17,8] and NGF [3] as well as the recently proposed approach based on global optimization of CMIF [10]. Figure 1 illustrates the general idea of the method.

A fast PyTorch-based implementation of the method is shared as open-source at http://github.com/MIDA-group/cross_sim_ngf.

## 2   Background

The (regularized) normalized gradient field [3], for image $A$ at point $x$, is

$$\vec{n}(x; A) = \frac{\nabla A(x)}{\sqrt{\|\nabla A(x)\|_2^2 + \epsilon^2}},\tag{1}$$

where $\epsilon$ is a small constant to reduce the impact of gradients with very small magnitude and avoid division by zero. In this work we use $\epsilon = 10^{-5}$ for $A(x) \in [0,1]$, selected empirically; higher values yielded more failed alignments and lower values mostly made the measure more noisy.

The main assumption of NGF-based alignment is that parts of images (acquired by different modalities) are in correspondence when the directions of their intensity changes are parallel or anti-parallel. A local similarity of NGF (SNGF) based on the squared dot-product of the elements of the NGF is defined as

$$s_{\mathrm{NGF}}(x; A, B) = \langle \vec{n}(x; A), \vec{n}(x; B) \rangle^2. \tag{2}$$

Orientation correlation (OC) and squared orientation correlation (SOC) offer an efficient way of computing SNGF of 2D images for all discrete displacements [1]. In 2D, the vectors $\vec{n}(\cdot; \cdot)$ are represented as complex numbers. A fast algorithm utilizing log-polar Fourier transform for OC-based alignment w.r.t. rotation and scaling is proposed in [16]. A computationally efficient extension to 3D [2] required a modification of the similarity measure; the authors proposed to, instead of (2), use its unsquared version:

$$s_{\mathrm{US\text{-}NGF}}(x; A, B) = \langle \vec{n}(x; A), \vec{n}(x; B) \rangle. \tag{3}$$

By observing three separable components of the (unsquared) dot-product in (3), the authors [2] formulated an algorithm for efficiently computing the measure for all discrete displacements using cross-correlation in the frequency domain. None of the existing work, however, describes a method for computing similarities of NGF using the squared measure (2) efficiently in the frequency domain for 3D volumes, a gap which we aim to fill with this work.

The ability to use the squared measure rather than the unsquared measure is beneficial for multimodal image alignment [1]. Eq. (3), similarly to (the unsquared) OC [1], exhibits useful properties such as invariance to changes of contrast and absolute intensity levels, which are suitable for monomodal registration tasks. However, multimodal scenarios are often characterized by the appearance of parts of a sample that are dark in one modality and bright in another; in such cases, aligned samples actually minimize $s_{\mathrm{US\text{-}NGF}}$.

## 3   Method

Here we define a similarity measure between NGF based on (2), a cross-similarity ($c.f.$ cross-correlation) formulation of the measure, and propose an algorithm for computing it efficiently in the frequency domain for all 3D discrete displacements.

In [3], the point-wise contributions of $s_{\mathrm{NGF}}$ (2) are aggregated by summation. A downside of this choice is that it imposes a strong bias towards full overlap of the images which can be especially problematic for global optimization. We instead formulate a scaled similarity measure that is applied to selected regions of the images $A\colon X_A \to \mathbb{R}$ and $B\colon X_B \to \mathbb{R}$, defined by indicator functions (masks) $M_A\colon X_A \to \{0,1\}$ and $M_B\colon X_B \to \{0,1\}$, ignoring the parts of the finite rectangular domains where either $M_A$ or $M_B$ are zero-valued. The average similarity of NGF is

$$s_{\mathrm{ANGF}}(A, B; M_A, M_B) = \frac{1}{\sum_x M_A(x) M_B(x)} \sum_x M_A(x) M_B(x) s_{\mathrm{NGF}}(x; A, B). \tag{4}$$

Based on $s_{\mathrm{ANGF}}$, we define the *Cross Similarity of NGF*

$$\mathrm{CSNGF}(\chi; A, B, M_A, M_B) = \frac{1}{N(\chi)} \sum_x M_A(x) M_B(x + \chi) s_{\mathrm{NGF}}(x; A(x), B(x+\chi)),$$
(5)

where $\chi \in S$ is a discrete translation from the set $S$ representing all the considered discrete translations and $N(\chi)$ is the number of overlapping voxels (where $M_A$ and $M_B$ intersect) as a function of $\chi$. $N(\chi)$ can be computed as the cross-correlation between the mask images $N(\chi) = (M_A \star M_B)(\chi)$. An analogous approach is taken in [10] to compute CMIF. Masks are essential for computation of CSNGF, for any choice of $S$ which results in a partial overlap of the images. Figure 1 illustrates CSNGF as a part of a rigid 3D alignment method.

A *direct method* for computing CSNGF for all $\chi \in S$ involves looping over each $\chi$, and compute and aggregate $s_{\mathrm{NGF}}$ for all overlapping voxels. If $|S| = O(|X_A|)$, then the run-time complexity of the direct method is $O(|X_A||X_B|)$ which for equisized images $A$ and $B$ gives a quadratic run-time complexity in the size of the images, which is not feasible for volumes of realistic sizes.

We propose a more efficient algorithm for computing CSNGF for all $\chi \in S$ in 3D. By reformulating (2), and expanding the squared dot-product,

$$s_{\mathrm{NGF}}(x; A, B) = \sum_{i=1}^{3} \left( \vec{n}_i(x; A)^2 \vec{n}_i(x; B)^2 + 2 \sum_{j=i+1}^{3} \vec{n}_i(x; A) \vec{n}_j(x; A) \vec{n}_i(x; B) \vec{n}_j(x; B) \right),$$
(6)

we express it as 6 separable parts comprising 3 squared components ($i \in \{1, 2, 3\}$), as well as products of 3 pairs of components ($(i, j) \in \{(1, 2), (1, 3), (2, 3)\}$), of the NGF vector fields (see Fig. 1a), which can be computed independently for all $\chi$ using cross-correlation. Let $\vec{n}_i^M$ denote a modified NGF scaled by the associated mask, $\vec{n}_i^M(x; A) = M_A(x) \vec{n}_i(x; A)$. The required cross-correlations $((\vec{n}_i^M(\cdot; A)^2) \star (\vec{n}_i^M(\cdot; B)^2))$ and $((\vec{n}_i^M(\cdot; A)\vec{n}_j^M(\cdot; A)) \star (\vec{n}_i^M(\cdot; B)\vec{n}_j^M(\cdot; B)))$ are efficiently computed in the frequency domain; $(\vec{n}_i^M(\cdot; A)^2 \star \vec{n}_i^M(\cdot; B)^2) = \mathcal{F}^{-1}\left(\overline{\mathcal{F}(\vec{n}_i^M(\cdot; A)^2)} \odot \mathcal{F}(\vec{n}_i^M(\cdot; B)^2)\right)$, where $\mathcal{F}(\cdot)$ denotes the Fourier transform, $\overline{z}$ denotes complex conjugation and $\odot$ denotes element-wise multiplication. For efficiency, the 6 separable parts are aggregated in the Fourier domain. Computing CSNGF involves 14 real-valued FFTs (6 per image plus 1 mask per image) and 2 inverse FFTs. Generalization to $n$D is straightforward.

### 3.1   Method for Global 3D Rigid Alignment

The fast algorithm for computing CSNGF for all $\chi \in S$ provides direct means of global optimization of $s_{\mathrm{ANGF}}$ w.r.t. axis-aligned shifts. To reach global optimization w.r.t. rigid transformations, we adopt a hybrid approach where the space of 3D rotations $\boldsymbol{\theta} = (\theta_x, \theta_y, \theta_z)$ (represented as Euler angles) is explored via a multi-stage combination of Gaussian pyramids, random search, and global optimization of $s_{\mathrm{ANGF}}$. One stage of this coarse-to-fine method is illustrated in Fig. 1. This multi-stage approach facilitates global search at the lowest considered resolution, followed by more local search to refine the alignment.

Initially, a Gaussian resolution pyramid with $m$ levels is constructed through the application of Gaussian blur and downsampling. For each level $k \in \{1 \ldots m\}$, a random search is performed in a coarse-to-fine sequence, by sampling angles $\boldsymbol{\theta}$ either (a) as random rotations from the set of all possible rotations, for the first level ($k = 1$), or (b) as rotations close to one of the $p_{k-1}$ best solutions of the previous level, for levels $k \in \{2, \ldots, m\}$. An angle "close to" is realized by perturbing the previous solution by a change in rotation around axes $(x, y, z)$, sampled from $\mathbb{U}(-u_{k-1}, u_{k-1})$ for each axis. For each $\boldsymbol{\theta}$, the corresponding transformation $T_{\boldsymbol{\theta}}$ is applied to the floating image $B_{\boldsymbol{\theta}} = B \circ T_{\boldsymbol{\theta}}$ using trilinear interpolation and its mask $M_{B_{\boldsymbol{\theta}}} = M_B \circ T_{\boldsymbol{\theta}}$ using nearest neighbor interpolation. $\vec{n}(\cdot; B_{\boldsymbol{\theta}})$ is computed, followed by computation of $\arg\max_{\chi} \mathrm{CSNGF}(\chi; A, B_{\boldsymbol{\theta}}, M_A, M_{B_{\boldsymbol{\theta}}})$ for all $\chi \in S$, where $S$ is the set of displacements satisfying a user-selected amount of minimum overlap $\gamma$. A suitable zero padding scheme is used to enable partial overlaps (following [10]). For $k > 1$, the $p_{k-1}$ best solutions of the previous level are also evaluated unmodified to not risk discarding good solutions. For $k = m$, the best rotation and displacement are taken as the final rigid transformation.

The method is parameterized by blur-levels $\sigma = (\sigma_1, \ldots, \sigma_m)$, downsampling factors $\mathbf{d} = (d_1, \ldots d_m)$, largest allowed steps $\mathbf{u} = (u_1, \ldots u_{m-1})$, number of rotations $\mathbf{a} = (a_1, \ldots a_m)$, and number of kept best solutions $\mathbf{p} = (p_1, \ldots, p_{m-1})$. For all related experiments, $\mathbf{d} = (4, 2, 2, 1)$, $\mathbf{a} = (5000, 3000, 300, 0)$, $\mathbf{u} = (10, 3, 0)$, and $\mathbf{p} = (20, 3, 1)$. We use $\gamma = 0.5$ everywhere in this study.

## 4 Performance Analysis

The empirical evaluation of the proposed method is based on the CERMEP-IDB-MRXFDG dataset [9], available upon request from the authors. The dataset consists of images of brains of 33 subjects acquired by 4 different modalities: T1 weighted MR, Flair MRI, Computed Tomography (CT), [18F] FDG PET, all mapped to the standard MNI space (see Fig. 1b), thus providing ground-truth for image alignment method evaluation, and a possibility to consider 6 different combinations of modalities, enabling evaluation of the generality of the methods.

### 4.1 Similarity landscape of the average SNGF

First, we perform an empirical analysis of how (4) is affected by spatial transformations of the observed images. The aim is to provide evidence of the relevance of global optimization for multimodal image alignment. We consider two images acquired with the modalities FLAIR and PET and study the similarity landscape as the PET volume is rotated around a single axis of rotation; the result is shown in Fig. 2. We observe that the similarity landscape exhibits characteristics that impede local methods without a good initial guess for all parameters.

### 4.2 Multimodal Brain Image Volume Alignment

We compare the proposed method with two global and two local alignment methods on the task of recovering rigid transformations of brain image volumes.

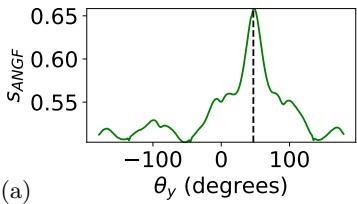 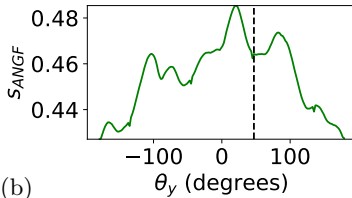

Fig. 2: Similarity landscape of $s_{\mathrm{ANGF}}$ for a pair of FLAIR and PET images of a brain (blur: $\sigma = 5$), w.r.t. the rotation angle $\theta_y$. Two scenarios are presented: (a) with no additional transformation, *i.e.*, all transformation parameters other than $\theta_y$ have their correct values, and (b) when the FLAIR image has been rotated by 5° around a random axis (other than $y$) and translated by 20 $vx$ in a randomly direction. The vertical dashed lines mark the sought angle. We observe that, (a) even without displacement, the convergence region of the sought angle has a limited size, with local maxima near the global maximum, and that (b) displacements along multiple dimensions make the search using local approaches further challenging; here the sought angle (dashed line) is between local optima.

For each of the twelve (ordered) pairs of modalities (six unordered modality combinations) included in the CERMEP-IDB-MRXFDG dataset, and for each of the first 20 subjects (the last 13 used for parameter tuning), we randomly (uniformly) sample a 3D rotation $\boldsymbol{\theta}$, and an axis-aligned shift $\chi_i \in [-30\ vx, +30\ vx]$ for each axis $i$. These transformations are applied, using inverse mapping and bicubic interpolation, to the first image volume of each pair. The transformed image is taken as reference image and the untransformed image as floating image in the alignment task. Finally, a block of size $151 \times 151 \times 151\ vx$ (*c.f.* original size $207 \times 243 \times 226$) at the center of the volume is extracted, retaining most of the content of interest, while omitting most of the background and avoiding padding introduced by inverse mapping outside the image domain. This setup enables evaluation of the accuracy of the proposed method w.r.t. alignment of multimodal 3D images by recovering these known transformations.

With the aim to evaluate the benefit of the proposed algorithm, based on the original similarity of NGF (2), compared to the one proposed in [2], we let USNGF refer to an alignment method similar to CSNGF, but with $s_{\mathrm{NGF}}$ in (5) replaced by $s_{\mathrm{US\text{-}NGF}}$. We evaluate both USNGF and "USNGF-", where the latter denotes USNGF but with an intensity-inverted floating image, to observe the sensitivity of USNGF to the sign of the gradients [1]. We also include the recently proposed CMIF-based global alignment method [10], which has exhibited excellent performance and outperformed several recent Deep Learning methods (including [12]) on multiple biomedical datasets. The selected global optimization methods are implemented in Python/PyTorch [11] with CUDA/GPU-acceleration.

We also compare with local optimization-based methods using MI and NGF as objective functions, relying on open-source implementations Elastix [8] and AIRLab [15] respectively.

We use the mean Euclidean distance between the corresponding corner points of the extracted block before and after the performed (recovered) alignment as a displacement measure, denoted $d_E$. We consider an alignment successful if $d_E < 5\ vx$, which is approximately 2% of the length of the diagonal of the blocks.

For CMIF we use $k = 16$ (for the $k$-means clustering), and $\sigma = (3.0, 1.5, 1.0, 0.0)$. For NGF, USNGF (and USNGF-), we use $\sigma = (5.0, 3.0, 2.0, 1.5)$. For local optimization MI (LO-MI) [17,8], we use 6 pyramid levels, the Adaptive Stochastic Gradient Descent optimizer [7], 4096 maximum iterations for each level. For local optimization NGF (LO-NGF) [3], we use 5 pyramid levels, ADAM optimizer, iteration counts according to the schedule (4096, 4096, 1024, 100, 50), with downsampling factors (16, 8, 4, 2, 1) and Gaussian smoothing parameters (15.0, 9.0, 5.0, 3.0, 1.0), with learning-rate 0.01. Trilinear interpolation is used.

**Results** The results of the evaluation of the 6 considered methods on the multimodal brain image dataset are presented in Tab. 1. The proposed method provides overall excellent performance, and is the best choice for all observed modality combinations. Most of the competitors show generally poor performance, completely failing on one or more modality combinations. Near-successes are also of interest, since those solutions may be refined with a local optimization method; therefore, we plot the distribution up to the threshold $d_E < 20$ as Fig. 3.

Table 1: Image alignment performance presented in terms of success-rate, where the threshold of success is set to $5\ vx$. The modality names are abbreviated in the headings (T: T1, F: Flair, C: CT, P: [18F] FDG PET).

| Method | T/F | T/C | T/P | F/C | F/P | C/P |
|---|---|---|---|---|---|---|
| LO-MI | 0.05 | 0.025 | 0.075 | 0.025 | 0.1 | 0.075 |
| LO-NGF | 0.025 | 0.00 | 0.00 | 0.00 | 0.00 | 0.00 |
| CMIF | 0.675 | 0.30 | 0.325 | 0.80 | 0.85 | 0.525 |
| USNGF | 0.225 | 0.00 | 0.00 | 0.00 | **0.925** | 0.10 |
| USNGF- | 0.00 | 0.275 | 0.00 | 0.00 | 0.00 | 0.00 |
| **CSNGF** | **1.00** | **0.95** | **0.925** | **0.90** | **0.925** | **0.95** |

### 4.3   Time Analysis

We compare the run-times of the global rigid registration methods, as well as the run-times of the novel Cross-Sim-NGF algorithm with a direct (not FFT-based) approach. The reported results are obtained on a Nvidia GeForce RTX 3090.

Both the FFT-based algorithm and the direct method are implemented in Python/PyTorch using GPU-acceleration; the direct method consists of a loop over all axis-aligned shifts $\chi \in S$, and computation of the squared dot-products.

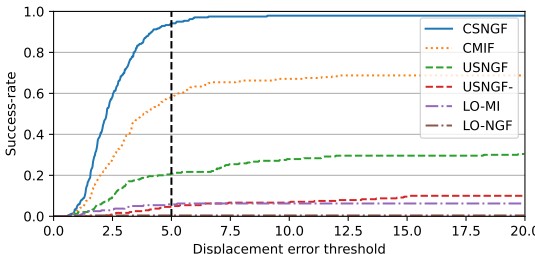

Fig. 3: Success-rate of each considered method as a function of the acceptable displacement error $t$ (fraction of the 240 alignments where $d_E < t$); the results for all modality combinations are aggregated. Up and to the left is better.

The average run-times of the methods CMIF, USNGF, and CSNGF are 569 s, 33 s, and 41 s, respectively. Comparison of the run-times of the FFT-based algorithm and the direct method, as a function of image size, is presented in Tab. 2. We observe that for size 128, the here proposed algorithm is approximately 6275 times (more than three orders of magnitude) faster.

Table 2: Run-time (s) comparison of FFT-based CSNGF and a direct algorithm for computing CSNGF, for all $\chi \in S$ where the overlap is 50% or higher, on cube image volumes of increasing size (expressed as side-length).

| Size
Method | 8 | 16 | 32 | 64 | 128 |
|---|---|---|---|---|---|
| Direct algorithm | 0.129 | 0.557 | 3.537 | 27.07 | 502.4 |
| **FFT-based alg.** | **0.002** | **0.002** | **0.002** | **0.008** | **0.088** |

## 5   Conclusion

We propose a novel NGF-based method for global rigid 3D multimodal alignment, which extends a well-performing method for 2D image alignment, outperforming a previous extension that relies on an unsquared version of the similarity measure. We confirm both its great performance and its high efficiency. Through the comparison with CMIF-based alignment [10], the method is indirectly compared with several approaches based on deep learning while leaving a more comprehensive comparative study as future work. The method does not use any training (data), which is a large advantage for (bio)medical applications [5].

## Acknowledgments

We thank Ines Mérida and team for the CERMEP-IDB-MRXFDG dataset. We acknowledge financial support by Vinnova, MedTech4Health Grant 2017-02447.

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
