

| | | | |
|---|---|---|---|
| (a) Ref1: PET | (b) Ref2: T1 | (c) Ref3: Flair | (d) Ref4: CT |
| (e) Flo1: T1 | (f) Flo2: Flair | (g) Flo3: CT | (h) Flo4: PET |
| (i) GT1: T1 | (j) GT2: Flair | (k) GT3: CT | (l) GT4: PET |

Fig. 4: Sample slices of 3D image pairs from the evaluation dataset generated from the CERMEP-IDB-MRXFDG dataset [9]. (a-d) the reference (transformed) images and (e-h) the floating images. Image (e) is to be registered to (a); (f) to (b), (g) to (c) and (h) to (d). The bottom row shows the ground-truth (GT) of each floating (Flo) image aligned to the corresponding reference (Ref) image.