# OpenReview forum: "Cross-Sim-NGF: FFT-Based Global Rigid Multimodal Alignment of Image Volumes using Normalized Gradient Fields"
_WBIR.info/2022/Workshop/Biomedical_Imaging_Registration — WBIR 2022_

### Official Review · Reviewer_Wd2H · 2022-02-14

**Rating:** 3
**Confidence:** 5
**Recommendation:** Short Oral

**Deanonymize Review:**

no

**Detailed Comments:**


1. In Table 1, LO-NGF measures attain zero success rate in all cases but one, which is hard to believe. Can you elaborate on the reasoning behind this?

2. Though, I agree with the author's claim that the local optimization schemes can be stuck at local optimal points. Fig 2. shows an example with smoothing factor $\sigma = 5.$. It seems that by further increasing the smoothing factor, the objection function will smooth out and the local scheme could attain a global optimal point. Do you agree?

3. Can we attain a good initial guess by choosing the multilevel strategy carefully for a given image pair for local optimization approaches? If yes, what is the advantage of your method?

4. Page 7, third paragraph: Why does each method used for comparison require a different level of smoothing (the value of sigma), downsampling factors, and optimization algorithms? For a fair comparison of methods, can we fix pyramid levels for all methods?

5. $vx$ denotes voxels?



**Paper Type:**

both

**Strengths Weaknesses:**

In the paper, the authors aim to register images with rigid transformation. They compute normalized gradient field (NGF) measures at numerous translation and rotation parameters and define an optimal parameter for which the NGF is maximum. They call their approach a global optimization approach.

The authors propose to compute normalized gradient field measures in the Fourier domain to reduce the overall computational time. The proposed technique works only with translation parameters. For rotation parameters, they follow the standard direct method.

Authors further show that their global optimization approach works better than existing local optimization approaches through experiments on a brain dataset.

The paper is overall well-written but lacks rigorous validation study of their method.

---

### Official Review · Reviewer_iPc9 · 2022-02-20

**Rating:** 2
**Confidence:** 5

**Deanonymize Review:**

no

**Detailed Comments:**

One problem with solving the registration problem in the frequency domain is that these methods are often very sensitive to noise for the calculation of rotation. These errors in rotation become more magnified as the distance from the point of rotation increases. The paper would be improved if the authors would address the sensitivity of their method to noise with respect to its effect on errors in rotation parameter estimation.

Fig 2b is not a meaningful illustration. There are three degrees of freedom for this example, but only one degree of freedom is shown. S_ANGF is a hyper surface in 4D space for this example and is therefore difficult to illustrate on paper. Fig 2b is not meaningful since it shows a slice through this hyper surface that does not include the global max.

**Paper Type:**

methodological development

**Strengths Weaknesses:**

The paper proposes an efficient algorithm for computing similarity of normalized gradient fields (NGF) in the frequency domain, which is globally optimized to achieve rigid multimodal 3D image alignment.

Strengths:
The authors propose to use normalized gradient fields to register multimodality images in the frequency domain. The authors solve this problem using exhaustive search approach and a GPU implementation.

The approach uses masks on the moving and target images to match only corresponding regions present in both images.

Weakness:

There is little novelty in this paper. The idea of solving for translation, rotation and scaling in the frequency domain has been around for a long time. Here is an early paper on the subject. A. Apicella, J.S. Kippenham, and J.H. Nagel, ‘‘Fast multi-modality image matching,’’ Proc. SPIE 1092, 252–263, 1989. Another paper on this subject is by Joseph Segman, Fourier cross correlation and invariance transformations for an optimal recognition of functions deformed by affine groups Vol. 9, No. 6, June 1992, J. Opt. Soc. Am. A.

The Methods section only explains how to compute the cost function, but does not provide any details on the transformation parameters.

There seems to be some mistakes in the method. The proposed method seems to suggest that the masks should be registered and not the images being masked. "N(χ) can be computed as the cross correlation between the mask images N(χ) = (MA ⋆ MB)(χ)." The correlation should be between the moving and target images and not the mask images. In the end, the authors seem to get to the correct equation since they multiply the mask by the images.

Another issue with the method is that the authors do not explain how they get the masks for the moving and target images. It is important for the masked region in the moving image to correspond to the masked region in the target image. The need for this discussion is further illustrated because the authors use the corners of the masks to evaluate their method. "We use the mean Euclidean distance between the corresponding corner points of the extracted block before and after the performed (recovered) alignment as a displacement measure, denoted d_E."

---

### Official Review · Reviewer_uSrh · 2022-02-21

**Rating:** 4
**Confidence:** 5

**Deanonymize Review:**

no

**Detailed Comments:**




* an fast algorithm --> a fast algorithm

**Paper Type:**

methodological development

**Strengths Weaknesses:**

* fast global multimodal registration tool; an efficient computation of NGF similarity of squared measure
* impressive differences in figure 1
* quite small experimental section
* What happens when the overlap between input images is minimal? Do no-overlap scenarios break the optimization process?
* experiments are run on a reasonably sized multi-modal data set
* modest innovation
* have the authors considered comparing theirs to "robust" tools in the field?(eg robust registration by M Reuter); it seems a bit strange how badly all other tools perform in the image alignment experiments
* newly proposed tool has good computational efficiency

---

### Official Review · Reviewer_LxAt · 2022-02-21

**Rating:** 4
**Confidence:** 5
**Recommendation:** Long Oral

**Deanonymize Review:**

yes

**Detailed Comments:**

I found the abbreviations, in particular  USNGF slightly unintuitive. Maybe you could use NGF^2 and NGF.  The capture range of the Fourier transform approach is in principle infinite (of course limited to the mask overlap as detailed in the paper), which is great but could also lead to problems if e.g. the field of view in one modality is cropped (see again CuRIOUS challenge). Could the authors elaborate on ways to limit the translation range for those cases? The search strategies for orientations/angles are rather conventional (random-search, multi-level etc.). Did the authors consider Powell's method as implemented in FMRIB's Flirt Toolbox ?
1. M. Jenkinson and S.M. Smith. A global optimisation method for robust affine registration of brain images. Medical Image Analysis, 5(2):143-156, 2001.
2. M. Jenkinson, P.R. Bannister, J.M. Brady, and S.M. Smith. Improved optimisation for the robust and accurate linear registration and motion correction of brain images. NeuroImage, 17(2):825-841, 2002.
Another idea that came into my mind: with a moderate computational overhead to a single global FFT, one could subdivide the image domain into a few overlapping patches and compute the displacement correlation independent for them. That way the contribution of different (large) regions could be separated and those "block-matching" results could be inserted into a closed form affine/rigid matrix solver, c.f. Aladin / NiftyReg
Block Matching algorithm for symmetric global registration. Based on Modat et al., “Global image registration using asymmetric block-matching approach” J. Med. Img. 1(2) 024003, 2014, doi: 10.1117/1.JMI.1.2.024003

**Paper Type:**

methodological development

**Strengths Weaknesses:**

Strengths:
- clear paper with good illustrations
- very interesting idea to decompose the inner-product computation of NGF and enable a global correlation (FFT) to find optimal translations
- reasonable sized evaluation dataset with different modality combinations and large (synthetic) misalignments
- source code in pytorch, which is also very clear and well-documented
Weaknesses:
- the local optimisation comparison produces surprisingly bad results
- the hyper parameter epsilon in Eq. 1 is not tuned per modality. To my knowledge setting it equally for different modalities does not always produce optimal results which is a known limitation of NGF
- more complex registration problems / benchmarks e.g. CuRIOUS (MRI-ultrasound brain alignment) could have been considered.

---

### Decision · Program_Chairs · 2022-02-22

Accept